# Fluorescence-Responsive Detection of Ag(I), Al(III), and Cr(III) Ions Using Cd(II) Based Pillared-Layer Frameworks

**DOI:** 10.3390/ijms24010369

**Published:** 2022-12-26

**Authors:** Qi-Jin Jiang, Po-Min Chuang, Jing-Yun Wu

**Affiliations:** Department of Applied Chemistry, National Chi Nan University, Nantou 545, Taiwan

**Keywords:** aluminum, chromium, fluorescence sensor, pillared-layer framework, silver

## Abstract

Two Cd(II) based coordination polymers, {Cd_3_(btc)_2_(BTD-bpy)_2_]∙1.5MeOH∙4H_2_O}_n_ (**1**) and [Cd_2_(1,4-ndc)_2_(BTD-bpy)_2_]_n_ (**2**), where BTD-bpy = bis(pyridin-4-yl)benzothiadiazole, btc = benzene-1,3,5-tricarboxylate, and 1,4-ndc = naphthalene-1,4-dicarboxylate, were hydro(solvo)thermally synthesized. Compound **1** has a three-dimensional non-interpenetrating pillared-bilayer open framework with sufficient free voids of 25.1%, which is simplified to show a topological (4,6,8)-connected net with the point symbol of (3^2^4^2^56)(3^4^4^4^5^4^6^2^8)(3^4^4^2^6^19^7^2^8). Compound **2** has a three-dimensional two-fold interpenetrating bipillared-layer condense framework regarded as a 6-connected primitive cubic (**pcu**) net topology. Compounds **1** and **2** both exhibited good water stability and high thermal stability approaching 350 °C. Upon excitation, compounds **1** and **2** both emitted blue light fluorescence at 471 and 479 nm, respectively, in solid state and at 457 and 446 nm, respectively, in the suspension phase of H_2_O. Moreover, compounds **1** and **2** in the suspension phase of H_2_O both exhibited a fluorescence quenching effect in sensing Ag^+^, attributed to framework collapse, and a fluorescence enhancement response in sensing Al^3+^ and Cr^3+^, ascribed to weak ion–framework interactions, with high selectivity and sensitivity and low detection limit.

## 1. Introduction

Metal and metal-containing materials are important resources widely used in many industries. Further, a large quantity of metal-containing products is also used in daily life to satisfy the requirements of human activities. However, improper discharge from industrial processes and unexpected leaching from improper treatment toward metal-wares would cause heavy metal ion pollution of the environment. Moreover, many heavy metal ions such as chromium, aluminum, silver, mercury, cadmium, lead, and iron are known to be very toxic or carcinogenic [1,2,3,4]. Short-term excessive heavy metal ion intake and long-term accumulation of heavy metal ions in the human body can result in toxic effects to human organisms, leading to numerous diseases [1,4,5,6,7]. For instance, Ag^+^ may cause cardiac enlargement, growth retardation, and degenerative changes in the liver [5,6]; Al^3+^ may be closely related to damage of the central nervous system [7,8]; Cr^3+^ may have impacts on the metabolism of carbohydrates, fats, proteins, and nucleic acids [1,8]. Therefore, it is unambiguous that heavy metal ion contamination is a serious global problem. As heavy metal ions represent severe risks for the aquatic and terrestrial ecosystems and even human survival and health [1,2,3,4,5,6,7,8,9], it is very important to sensitively and selectively detect trace amounts of heavy metal ions in aqueous solution for human health monitoring and environmental protection.

The traditional instrumental analytical techniques such as atomic absorption spectrometry (AAS) [10,11], inductively coupled plasma mass spectrometry/optical atomic emission spectrometry (ICP-MS/OES) [12,13,14], atomic fluorescence spectrometry (AFS) [15], X-ray fluorescence spectrometry (XRF) [16], and electrochemical methods [17,18], are widely used for the determination of environmental pollutants, especially heavy metal ions, with high accuracy, good sensitivity, and low detection limits. However, these analytical techniques are usually limited by expensive instruments, time-consuming sample pretreatment, well-controlled experimental conditions, and high cost, and they require well-trained operators [19,20,21,22]. Therefore, there is an urgent need to develop inexpensive and accurate detection methods, such as fluorometric approaches, to have the advantages of facilitated detection, easy manipulation, rapid response, and low cost for the rapid and effective determination of trace amounts of chemical pollutants such as metal ions in various media [21,22,23].

Luminescent coordination polymers (CPs) and luminescent metal–organic frameworks (MOFs) recently have emerged as a kind of luminescence sensing materials to be reckoned with, owing to their rich and fascinating luminescence properties [24,25,26]. A large number of studies have been concerned with the construction of novel luminescent CPs/MOFs for applications in the effective detection of hazardous pollutants, such as heavy metal ions, by enhancing or quenching fluorescence in intensity and/or shifting fluorescence in wavelength upon stimulation [19,20,21,22,23,27,28,29,30,31,32,33,34,35,36,37,38,39,40,41,42,43,44,45]. Subsequent to our efforts on luminescence detection using luminescent CPs/MOFs as potential multifunctional sensing platforms [46,47,48], herein, two cadmium(II) based luminescent CPs, {Cd_3_(btc)_2_(BTD-bpy)_2_]∙1.5MeOH∙4H_2_O}_n_ (**1**, H_3_btc = benzene-1,3,5-tricarboxylic acid, BTD-bpy = bis(pyridin-4-yl)benzothiadiazole, Figure 1) and [Cd_2_(1,4-ndc)_2_(BTD-bpy)_2_]_n_ (**2**, 1,4-H_2_ndc = naphthalene-1,4-dicarboxylic acid), were hydro(solvo)thermally synthesized. Compounds **1** and **2** both adopt three-dimensional pillared-layer frameworks featuring the non-interpenetrating (4,6,8)-connected net and two-fold interpenetrating 6-connected net, respectively. Compounds **1** and **2** both emitted blue light emissions and thus were exploited as fluorescence-responsive sensor platforms for Ag^+^ detection through the fluorescence quenching effect and Al^3+^ and Cr^3+^ detection through fluorescence enhancement response with high selectivity and sensitivity in H_2_O. Possible sensing mechanisms were also investigated.

## 2. Results and Discussion

### 2.1. Crystal Structure Description of {[Cd_3_(btc)_2_(BTD-bpy)_2_]∙1.5MeOH∙4H_2_O}_n_ (**1**)

Compound **1** crystallized in the monoclinic space group *P*2/*c*, and the asymmetric unit contains a one and half Cd(II) center, one btc^3−^ ligand, and one BTD-bpy ligand. The Cd(II) at general position, labeled as Cd(1), is coordinated by five O atoms from three different btc^3−^ ligands (Cd−O = 2.289(5)−2.585(6) Å) and two N atoms from two BTD-bpy ligands (Cd−N = 2.278 (5) − 2.305 (6) Å), furnishing a distorted {CdO_5_N_2_} pentagonal bipyrimidal geometry, where the five O atoms are located at the equatorial plane and the two N atoms are placed at the axial positions (Figure 1a). The Cd(II) at special position passing through 2-fold rotation axis, labeled as Cd(2), is coordinated by six O atoms from four different btc^3−^ ligands (Cd−O = 2.313 (7) − 2.382 (7) Å), furnishing a distorted {CdO_6_} trigonal antiprism geometry (Figure 1b). Each btc^3–^ ligand links five Cd(II) centers, where one carboxylate group is in a chelating (*κ*^2^O,Oʹ) coordination mode and the other two carboxylate groups both are in a bidentate chelating-bridging (*μ*_2_-*κ*^2^O,Oʹ:*κ*^1^Oʹ) coordination mode (Figure 1c).

Structural analysis indicated that the Cd(II) centers are connected by btc^3−^ ligands to form a two-dimensional bilayer structure of [Cd_3_(btc)_2_]*_n_* extending in the ac plane (Appendix A). Two Cd(1) centers are *μ*_2_-*κ*^2^O,Oʹ:*κ*^1^Oʹ-bridged by two carboxylate groups of two different btc^3−^ ligands and each of them is further *κ*^2^O,Oʹ-chelated by one carboxylate group from a further btc^3−^ ligand, forming an edge-sharing dinuclear secondary building unit (SBU) of {Cd(1)_2_(*μ*_2_-CO_2_)_2_(CO_2_)_2_} with the Cd(1)∙∙∙Cd(1) separation of 3.8897 Å (Figure 1a). The dinuclear {Cd(1)_2_(*μ*_2_-CO_2_)_2_(CO_2_)_2_} SBUs are shared corners with the mononuclear {Cd(2)(CO_2_)_4_} units to result in one-dimensional {Cd_3_(CO_2_)_6_}*_n_* chains running along the [100] direction with a Cd(1)∙∙∙Cd(2) separation of 4.5240 Å (Appendix A). Topologically, the dinuclear {Cd(1)_2_(*μ*_2_-CO_2_)_2_(CO_2_)_2_} SBU and the mononuclear {Cd(2)(CO_2_)_4_} unit both can be regarded as 6-connected nodes and the btc^3−^ ligand can be regarded as a 4-connected node (Figure 1d–f). Therefore, the bilayer structure is simplified as a (4,6,6)-connected net with the point symbol of (3^2^4^2^56)(3^4^4^4^5^4^6^2^8)(3^4^4^2^6^7^7^2^) (Appendix A). Noteworthy, the BTD-bpy ligands act as bis(monodentate) pillars to connect only the dinuclear {Cd(1)_2_(*μ*_2_-CO_2_)_2_(CO_2_)_2_} units in neighboring [Cd_3_(btc)_2_]*_n_* bilayers, generating a three-dimensional open framework having sufficient free voids of 25.1% per unit cell volume calculated by PLATON (Figure 1g,h) [49]. From the topological point of view, such a pillared-bilayer framework is regarded as a (4,6,8)-connected net with the point symbol of (3^2^4^2^56)(3^4^4^4^5^4^6^2^8)(3^4^4^2^6^19^7^2^8) (Figure 1i), where the dinuclear {Cd(1)_2_(*μ*_2_-CO_2_)_2_(CO_2_)_2_} unit is expanding to be an 8-connected node (Figure 1d).

### 2.2. Crystal Structure Description of [Cd_2_(1,4-ndc)_2_(BTD-bpy)_2_]_n_ (**2**)

Compound **2** crystallizes in the monoclinic space group *P*2_1_/*c*, and the asymmetric unit contains two Cd(II) center, two 1,4-ndc^2−^ ligands, and two BTD-bpy ligands. Each Cd(II) is coordinated by four O atoms from three different 1,4-ndc^2–^ ligands (Cd−O = 2.2336 (19) − 2.406 (2) Å) and two N atoms from two BTD-bpy ligands (Cd−N = 2.348 (2) − 2.400 (2) Å), furnishing a distorted {CdO_4_N_2_} octahedral geometry (Figure 2a), where the four O atoms are located at the equatorial plane and the two N atoms are placed at the axial positions. Each 1,4-ndc^2−^ ligand links three Cd(II) centers, where one carboxylate group is in a chelating (*κ*^2^O,Oʹ) coordination mode and the other carboxylate group is in a bidentate bridging (*μ*_2_-*κ*^1^O:*κ*^1^Oʹ) coordination mode (Figure 2b). Two Cd(II) centers are *μ*_2_-*κ*^1^O:*κ*^1^Oʹ-bridged by two carboxylate groups of two different 1,4-ndc^2−^ ligands and each of them is further *κ*^2^O,Oʹ-chelated by one carboxylate group from a further 1,4-ndc^2−^ ligand, forming a dinuclear SBU of {Cd_2_(*μ*_2_-CO_2_)_2_(CO_2_)_2_} with the Cd∙∙∙Cd separation of 4.4508 Å, where the four remaining axial positions are occupied by BTD-bpy ligands (Figure 2c). Topologically, the dinuclear unit can be considered as a 6-connected node (Figure 2d). Structural analysis indicated that each {Cd_2_(*μ*_2_-CO_2_)_2_(CO_2_)_2_} unit is connected by four identities to form a two-dimensional layer structure of [Cd_2_(1,4-ndc)_2_]*_n_* showing the square lattice (**sql**) topology (Figure 2e). The grid has rhombic windows with node-to-node distances of 12.607 Å × 16.175 Å in edge. The [Cd_2_(1,4-ndc)_2_]*_n_* layers are pillared by bis(monodentate) BTD-bpy ligands to govern a three-dimensional bipillared-layer open framework, which is simplified to be a topological primitive cubic (**pcu**) net (Figure 2f). Noteworthy, the bipillared-layer frameworks are mutually interpenetrated in 2-fold degrees (Figure 2g), greatly reducing the sufficient free voids to only about 0.8% per unit cell volume [49].

### 2.3. X-ray Powder Diffraction (XRPD) and Thermogravimetric (TG) Analysis

The phase purity of the bulky samples of **1** and **2** were validated by XRPD measurements undoubtedly, which showed well-matched X-ray diffractograms between the experimental and simulated XRPD patterns (Figure 3). Further, the XRPD patterns of the recovered samples after immersing in water for 1 day were in agreement with the pristine samples (Figure 3), confirming the water stability of **1** and **2**.

Thermogravimetric (TG) analysis of **1** conducted under a nitrogen atmosphere revealed a gradual weight loss upon heating from room temperature to approximately 140 °C (Appendix A), corresponding to the escape of lattice MeOH and H_2_O molecules (found 7.3%; calcd. 8.3% based on Cd_3_(btc)_2_(BTD-bpy)_2_]∙1.5MeOH∙4H_2_O). The coordination framework of **1** was thermally stable upon heating to approximately 350 °C, which was then decomposed to leave CdO as final residues (found 25.5%; calcd. 26.5%). The TG curve of **2** showed a large plateau from room temperature to approximately 350 °C, implying its high thermal stability. The framework was then decomposed to leave CdO as final residues (found 19.4%; calcd. 20.8%).

### 2.4. Gas Adsorption Properties

As compound **1** has 25.1% potential free void per unit cell volume, its gas adsorption properties after thermal activation were examined. For thermally activated **1** at P/P_0_ = 1, the gas adsorption isotherms showed only 5.19 cm^3^ g^−1^ for N_2_ uptake at 77 K and 6.92 cm^3^ g^−1^ for CO_2_ uptake at 195 K (Appendix A). The low N_2_ uptakes gave the estimation of Brunauer−Emmett−Teller (BET) surface areas of 2.5 m^2^ g^−1^ (the estimated Langmuir surface area: 6.6 m^2^ g^−1^).

After N_2_ and CO_2_ adsorption−desorption, the XRPD patterns of **1** were measured. As observation, though there was some apparent peak broadening and peak shifting, the checked XRPD patterns were very similar to that of the as-synthesized/simulated profiles of **1** (Appendix A). This implies quasi-closed windows for N_2_ and CO_2_, resulting in the occurrence of only surface adsorption.

### 2.5. Photoluminescent Properties

The photoluminescent properties of BTD-bpy, H_3_btc, and 1,4-H_2_ndc ligands and compounds **1** and **2** were studied at room temperature in solid state and in H_2_O (Appendix A). Upon excitation, BTD-bpy, H_3_btc, and 1,4-H_2_ndc all showed emission band(s), centered at 477 (*λ*_ex_ = 335 nm), 384 (*λ*_ex_ = 320 nm), and 476 nm (*λ*_ex_ = 365 nm), respectively, in solid state, and at 448 (*λ*_ex_ = 315 nm), 342/397 (*λ*_ex_ = 270 nm), and 440 nm (*λ*_ex_ = 325 nm), respectively, in H_2_O. For **1** and **2**, both of them exhibited a fluorescence emission band at around 471 and 479 nm, respectively, in solid state and at 457 and 446 nm, respectively, in H_2_O suspensions upon excitation. These bands resembled the emission band of the BTD-bpy ligand and that of the free 1,4-H_2_ndc ligand very closely but differed from that of free H_3_btc. Therefore, the emission band of **1** mainly originated from the intraligand (IL) transitions of the BTD-bpy ligand and that of **2** can be ascribed to the combination of the IL transitions of BTD-bpy and 1,4-ndc^2−^ ligands upon coordination.

### 2.6. Fluorescence Sensing of Metal Ions

Sensing experiments were performed for 1 mg of each complex in 3 mL of H_2_O. The well-prepared suspensions of **1** and **2** in H_2_O exhibited particle sizes of 372.3 ± 102.9 nm and 362.2 ± 100.2 nm, respectively (Appendix A), suggesting uniformity. In addition, the measured zeta potentials of **1** and **2** are −8.41 and 1.46 mV, respectively, in the natural pH conditions, implying negatively and positively charged crystal surfaces, respectively.

Compounds **1** and **2** in the suspension phase of H_2_O both emitted blue light emissions, making them potential candidates to be exploited as fluorescence-responsive sensing platforms. Therefore, the fluorescence sensing performances of **1** and **2** toward metal ions were investigated at room temperature. Metal ions including Ag^+^, Al^3+^, Ca^2+^, Cd^2+^, Co^2+^, Cr^3+^, Cu^2+^, K^+^, Na^+^, Mg^2+^, Mn^2+^, Ni^2+^, and Pb^2+^ ions were chosen as analytes. After being uniformly dispersed in H_2_O, the emission spectra of **1** and **2** were measured before and after the addition of each of the metal ion analytes with a concentration of 1 mM. As shown in Figure 4, most metal ions involving Ca^2+^, Cd^2+^, Co^2+^, K^+^, Na^+^, Mg^2+^, Mn^2+^, Ni^2+^, and Pb^2+^ ions displayed negligible effects in changing the emission intensity. The coinage metal ions such as Ag^+^ and Cu^2+^ ions caused significant fluorescence quenching effects, especially Ag^+^. The fluorescence quenching percentages are ca. 80 and 90% for **1** and **2**, respectively, by the Ag^+^ ion and ca. 50 and 22% for **1** and **2**, respectively, by the Cu^2+^ ion. On the contrary, the Al^3+^ and Cr^3+^ ions demonstrated remarkable fluorescence enhancement responses. The fluorescence enhancement ratios are ca. 2.0 and 3.1 times for **1** and **2**, respectively, by the Al^3+^ ion and ca. 2.3 and 2.9 times for **1** and **2**, respectively, by the Cr^3+^ ion. As a result, both **1** and **2** could be promising fluorescence-responsive sensor platforms for detecting Ag^+^, Al^3+^, and Cr^3+^ ions in H_2_O. Comparably, the emission of BTD-bpy in H_2_O was quenched after the addition of Ag^+^, while it was enhanced after the addition of Al^3+^ and Cr^3+^ ions (Appendix A); the changes were very close to those of **1** and **2** with the addition of Ag^+^, Al^3+^, and Cr^3+^ in H_2_O. On the other hand, the emission spectra of H_3_btc and 1,4-H_2_ndc showed some changes in intensity with the addition of Ag^+^, Al^3+^, and Cr^3+^ in H_2_O (Appendix A). These findings indicate that Ag^+^, Al^3+^, and Cr^3+^ ions favorably formed analyte−sensor interactions with BTD-bpy struts rather than btc^3−^ and 1,4-ndc^2−^ in **1** and **2** during sensing.

The competitive detection experiments were carried out to examine the potential of **1** and **2** as an excellent fluorescence sensor showing high selectivity. Experimental results indicated that **1** and **2** both displayed excellent anti-interference ability for the highly selective detection of Ag^+^ due to the negligible changes in fluorescence intensity perturbed by competitive ions (Figure 5). Comparably, in cases of Al^3+^ and Cr^3+^ detection, interference from Ag^+^ ions was apparently observed for both **1** and **2,** whereas Al^3+^ and Cr^3+^ competed for recognition of **1** and **2**. Other metal ion analytes had no significant influence in changing the fluorescence intensities of **1** and **2**. These results clearly implied that **1** and **2** both had extremely high recognition ability to distinguish Ag^+^ ions from other metal ions, and acceptable anti-interference ability to selectively detect Al^3+^ and Cr^3+^ ions under specific controlled conditions, for example, in the absence of Ag^+^ ions.

To examine the fluorescence sensing sensitivity, the quantitative titration experiments of **1** and **2** in the suspension phase of H_2_O were conducted with the incremental addition of analyte concentration involving Ag^+^, Al^3+^, and Cr^3+^ ions, and the related fluorescence emission spectra were measured. In the case of Ag^+^ detection, the fluorescence intensities of **1** and **2** both were gradually decreased as the Ag^+^ concentration was regularly increased from 0 to 0.10 mM for **1** and from 0 to 1.0 mM for **2** (Figure 6). Noteworthy, **1** and **2** both displayed significant fluorescence change in intensity but not in wavelength, as the fluorescence bands of **1** and **2** both are almost un-shifted in wavelength during the whole titration concentrations. The Stern−Volmer equation, *I*_0_/*I* = *K*_SV_ × [Quencher] + 1, was utilized to analyze the detection sensitivity, whereas the equation, DL = 3*σ*/*k*, where *σ* denotes the standard deviation of blank measurements and *k* denotes the absolute value of the slope, was applied to determine the detection limit. For the quantitative detection of Ag^+^ ions, the Stern−Volmer plot of [(*I*_0_/*I*) − 1] against Ag^+^ concentration gave a good linear relationship in the range of 0−0.10 mM with the correlation coefficient *R*^2^ = 0.9973 for **1** and 0–1.0 mM with the correlation coefficient *R*^2^ = 0.99649 for **2**, and accordingly, the *K*_SV_ was determined to be 2.053 × 10^4^ M^−1^ for **1** and 1.598 × 10^4^ M^−1^ for **2** (insets in Figure 6). Further, the DL value was calculated to be 0.56 μM for **1** and 1.47 μM for **2** toward Ag^+^ detection. A series of CP/MOF-based fluorescence sensors applied in Ag^+^ detection are surveyed in Appendix A. As observation, the DL values of **1** and **2** for Ag^+^ sensing are on the same order of magnitude as other reported luminescent CP/MOF-based probes [20,23,27,28,50,51,52,53]. Therefore, **1** and **2**, especially the former, possess remarkable sensing competitiveness for Ag^+^ detection via the fluorescence ON−OFF mechanism.

In cases of Al^3+^ and Cr^3+^ detection, on the other hand, when the Al^3+^ or Cr^3+^ concentrations were incrementally added, the fluorescence intensities of **1** and **2** both were continuously enhanced (Figure 7), suggesting diffusion-controlled fluorescence enhancement. Further, the fluorescence bands of **1** and **2** both are gradually blue-shifted in wavelength by 21 ± 1 nm for Al^3+^ detection and 22 ± 2 nm for Cr^3+^ detection during the whole titration concentrations. Of particular note, the fluorescence intensity and the analyte (Al^3+^ or Cr^3+^) concentration fit the first-order exponential decay relationship very well, with the formulae of *I* = −297.28 × exp(−[Al^3+^]/0.062) + 829.26 (*R*^2^ = 0.99508) for Al^3+^ detection and *I* = −814.37 × exp([Cr^3+^]/0.31) + 1441.26 (*R*^2^ = 0.99861) for Cr^3+^ detection by **1**, and *I* = −1224.32 × exp(−[Al^3+^]/0.30) + 1722.25 (*R*^2^ = 0.99848) for Al^3+^ detection and *I* = −1441.27 × exp(−[Cr^3+^]/0.18) + 1853.56 (*R*^2^ = 0.99792) for Cr^3+^ detection by **2**. In addition, the DL value was calculated to be 4.97 μM for **1** and 0.25 μM for **2** toward Al^3+^ detection and 3.03 μM for **1** and 0.79 μM for **2** toward Cr^3+^ detection. Appendix A display a comparison of literature reports for Al^3+^ and Cr^3+^ detection, respectively, using CP/MOF-based fluorescence sensors in water [22,29,30,31,32,33,34,35,36,37,38,39,40,41,42,43]. As a representative, **1** and **2** display comparable sensing performances toward Al^3+^ and Cr^3+^. Of particular note, **2** can be an effective fluorescence OFF−ON sensor for Al^3+^ and Cr^3+^, owing to the lower detection limit.

### 2.7. Sensing Mechanism

The plausible sensing mechanisms were explicated. After being immersed in Ag^+^ aqueous solutions for 1 day, the recovery solids of **1** and **2** both revealed amorphous XRPD profiles (Figure 3), which is an indication of the loss of crystallinity. This vividly implies the correlation between fluorescence turn-off and architecture collapse. In other words, framework collapse is the most likely interpretation for the fluorescence turn-off sensing toward Ag^+^ by **1** and **2** in the suspension phase of H_2_O. This is compared with other reports on Ag^+^ detection based on quenching effects, which is dominated by a charge transfer process between Ag^+^ and a fluorescence-quenched chemosensor [53,54,55].

In cases of luminescence enhancement sensing of Al^3+^ and Cr^3+^ by **1** and **2** in the suspension phase of H_2_O, the recovery solids of **1** and **2** immersed in Al^3+^ or Cr^3+^ aqueous solutions for 1 day all displayed similar XRPD patterns to that of pristine **1** and **2,** with no significant changes in the diffraction positions and intensity (Figure 3). This indicates that the structure and crystallinity of **1** and **2** retained integrity after ion sensing in water, proving the high structure stability during sensing again. In addition, the possibility that structure disruption caused fluorescence turn-on detection can be eliminated.

The literature has shown that most reports on metal ion detection based on enhancement effects are due to the weak interaction between the ionic analytes and framework [29,30,31,32,33,34,42,43]. Therefore, the IR and XPS spectra of **1** and **2** were explored to verify this assumption. After immersing **1** in Al^3+^ and Cr^3+^ aqueous solutions for 1 day, the IR spectra of recovery samples showed no significant changes compared with that of as-synthesized **1** (Appendix A), suggesting no or an extremely weak framework–M^3+^ (M^3+^ = Al^3+^, Cr^3+^) interactions that cannot cause significant changes in IR vibrations.

However, the IR spectra of **2** before and after being treated with Al^3+^ and Cr^3+^ in H_2_O for 1 day displayed remarkable differences (Appendix A). The antisymmetric and symmetric stretching vibrations of carboxylate group (RCOO^−^) at 1550 and 1360 cm^−1^, respectively, in **2** were largely reduced in intensity after treatment with Al^3+^ and Cr^3+^. The weak stretching vibration of the benzothiadiazole C=N bond at 1704 cm^−1^ was red-shifted to 1690 cm^−1^ as a strong peak in both Al^3+^- and Cr^3+^-treated **2**. In addition, the band at 1412 cm^−1^, corresponding to the stretching vibration of the benzothiadiazole C=C bond, was slightly blue-shifted by 11 cm^−1^. The differences in the IR spectra of **2** after treatment with Al^3+^ and Cr^3+^ in H_2_O pointed to the framework−M^3+^ (M^3+^ = Al^3+^, Cr^3+^) interactions that were mainly occupied at the benzothiadiazole N positions and, in part, at the 1,4-ndc^2−^ O positions.

XPS spectra provided further support (Figure 8). The XPS spectra of **1** and **2** after being treated with Al^3+^ and Cr^3+^ in H_2_O for 1 day showed significant shifts in binding energy in the O 1*s* peak, which gave the binding energy of 531.6 eV for as-synthesized **1** and 532.2 and 531.9 eV for Al^3+^- and Cr^3+^-treated **1**, respectively, as well as 528.8/531.8 eV for as-synthesized **2** and 527.0/532.6 and 528.4/532.3 eV for Al^3+^- and Cr^3+^-treated **2**, respectively, implying the formation of framework−M^3+^ (M^3+^ = Al^3+^, Cr^3+^) interactions. Moreover, the shifts in O 1s in **2** is obviously larger than that in **1**, reflecting stronger framework−M^3+^ (M^3+^ = Al^3+^, Cr^3+^) interactions. In addition, the N 1*s* peak is divided into two peaks, where the component at around 400 eV is typical of the pyridyl group [56,57], and the other at around 406 eV can be attributed to the benzothiadiazole group. The former at 399.7 and 399.9 eV for as-synthesized **1** and **2**, respectively, showed very small shifts (≤0.3 eV) after being treated with Al^3+^ and Cr^3+^ in water, while the latter at 405.2 and 405.3 eV for as-synthesized **1** and **2**, respectively, displayed larger shifts (ca. 0.3–1.4 eV) after being treated with Al^3+^ and Cr^3+^ in H_2_O. This implies that the benzothiadiazole group also formed contacts with Al^3+^ and Cr^3+^ ions through the N donors. The S 2*p*_3/2_ peak appeared at around 166 eV in the XPS spectra of **1,** and **2** displayed almost unchanged shifts before and after treatment with Al^3+^ and Cr^3+^ in H_2_O.

Structural analyses showed that **1** has large pores of about 25.1% while **2** possesses very small pores of about 0.8%. The difference in pore size makes metal ions pass through the framework more easily in **1** compared to **2**, resulting in weaker framework–M^3+^ (M^3+^ = Al^3+^, Cr^3+^) interactions. In addition, the benzothiadiazole and the 1,4-ndc^2−^ in **2** are close to each other in the pores, creating a comfortable region for the binding of metal ion analytes to form stronger framework–M^3+^ (M^3+^ = Al^3+^, Cr^3+^) interactions, and such an environment is not observed in **1** since the benzothiadiazole and the btc^3−^ are not close to each other. This is consistent with the results from IR and XPS experiments.

In whole, these experimental observations imply weak interactions between the Al^3+^/Cr^3+^ ions and frameworks (**1** and **2**), which would facilitate energy transfer and result in intraligand (IL) transitions of the BTD-bpy ligand, leading to luminescence enhancement.

## 3. Materials and Methods 

### 3.1. Chemicals and Instruments

Chemicals were commercially obtained and used without further purification. Ligand BTD-bpy was prepared according to the literature report [58]. Infrared (IR) spectra were carried out on a Perkin-Elmer Frontier Fourier transform infrared spectrometer (Perkin-Elmer, Taipei, Taiwan) using the attenuated total reflection (ATR) sampling technique. Thermogravimetric (TG) analyses were recorded on a Thermo Cahn VersaTherm HS TG analyzer (Thermo, Newington, NH, USA) under a flow of nitrogen atmosphere. X-ray powder diffraction (XRPD) measurements were conducted on a Shimadzu XRD-7000 diffractometer (Shimadzu, Kyoto, Japan) using graphite monochromatized Cu Kα radiation (λ = 1.5406 Å) set at 40 kV and 30 mA. Fluorescence emission spectra were carried out on a Hitachi F2700 fluorescence spectrophotometer (Hitachi, Tokyo, Japan) at room temperature using a 150 W xenon lamp as an excitation source. UV-vis absorption spectra were performed on a JASCO V-750 UV/VIS spectrophotometer (JASCO, Tokyo, Japan). Elementary microanalyses were performed on an Elementar Vario EL III analytical instrument (Elementar, Langenselbold, Germany). X-ray photoelectron spectroscopy (XPS) analyses were conducted on a Thermo Scientific ESCALAB 250 surface analysis system (Thermo Fisher Scientific Inc., Waltham, MA, USA) using a monochromatic Al Kα source. The C1’*s* core level of adventitious carbon at 284.6 eV was set as the reference for binding energy calibration. Zeta potentials and particle size distribution were analyzed from the experimental data recorded on a Micromeritics NanoPlus-3 Zeta Potential/Nano Particle analyzer (Micromeritics Instrument Corporation, GA, USA).

### 3.2. Synthesis of {Cd_3_(btc)_2_(BTD-bpy)_2_]∙1.5MeOH∙4H_2_O}_n_ (**1**)

A methanol solution (3 mL) of BTD-bpy (7.3 mg, 0.025 mmol), an aqueous solution (1 mL) of Cd(NO_3_)_2_∙4H_2_O (11.8 mg, 0.050 mmol), and a methanol solution (1 mL) of benzene-1,3,5-tricarboxylic acid (H_3_btc, 10.5 mg, 0.050 mmol) were sequentially sealed in a Teflon-lined autoclave. The autoclave was heated to 120 °C from room temperature in a period of 6 h, held at that temperature for 48 h, and then cooled to 30 °C in a period of 36 h. The resultant yellow crystals were collected by filtration, washed with methanol, and dried at room temperature. Yield: 80% based on BTD-bpy (13.4 mg, 0.010 mmol). IR (ATR, cm^−1^): 1697, 1606, 1542, 1431, 1367, 1296, 1218, 1108, 1069, 1016, 926, 887, 816, 757, 725. Found: C, 43.05; H, 2.52; N, 7.56%. Anal. Calcd Required for C_50_H_26_Cd_3_N_8_O_12_S_2_∙4H_2_O: C, 42.73; H, 2.42; N, 7.80%.

### 3.3. Synthesis of [Cd_2_(1,4-ndc)_2_(BTD-bpy)_2_]_n_ (**2**)

A methanol solution (3 mL) of BTD-bpy (7.3 mg, 0.025 mmol), an aqueous solution (1 mL) of Cd(NO_3_)_2_∙4H_2_O (11.8 mg, 0.050 mmol), and a methanol solution (1 mL) of naphthalene-1,4-dicarboxylic acid (1,4-H_2_ndc, 10.5 mg, 0.050 mmol) were sequentially sealed in a Teflon-lined autoclave. The autoclave was heated to 120 °C from room temperature in a period of 6 h, held at that temperature for 48 h, and then cooled to 30 °C in a period of 36 h. The resultant yellow crystals were collected by filtration, washed with methanol, and dried at room temperature. Yield: 80% based on BTD-bpy (12.9 mg, 0.010 mmol). IR (ATR, cm^−1^): 3043, 1704, 1550, 1412, 1360, 1085, 1049, 1013, 915, 820, 600. Elem. microanal. for C_56_H_32_Cd_2_N_8_O_8_S_2_: C, 54.51; H, 2.61; N, 9.08%. Found: C, 54.20; H, 2.65; N, 8.99%.

### 3.4. Single-Crystal X-ray Structure Determination

Single-crystal X-ray diffraction analyses were performed on a Bruker D8 Venture diffractometer using a graphite monochromatized Mo K*α* radiation (*λ* = 0.71073 Å). The structures were solved using the SHELXS-97 program [59] and refined based on *F*^2^ by full-matrix least-squares methods using the SHELXL-2014/7 [60], incorporated in WINGX-v2014.1 crystallographic collective package [61]. Non-hydrogen atoms were refined anisotropically while hydrogen atoms were refined isotropically. For **1**, the lattice solvent molecules are highly disordered and cannot be properly modeled. The SQUEEZE instruction thereby was conducted to remove the diffraction contributions from these lattice solvent molecules [62]. The calculations indicated the presence of 286 electrons per unit, which was associated with the thermogravimetric (TG) analysis and elementary microanalysis to suggest approximately 1.5 MeOH and 4 H_2_O molecules per formula. Experimental details for X-ray data collection and the refinements are summarized in Table 1. CCDC 2213904 (**1**) and 2213905 (**2**) contain the supplementary crystallographic data for this paper. These data can be obtained free of charge from the Cambridge Crystallographic Data Centre via www.ccdc.cam.ac.uk/data_request/cif.

### 3.5. Fluorescence Sensing Measurements

The fluorescence sensing measurements were performed in H_2_O suspensions at room temperature. The fine-ground crystalline sample (1 mg) was immersed in H_2_O (3 mL) and then ultrasonically agitated with pulsed ultrasound for 10 min to form homogeneously stable suspensions. The aqueous solutions (0.10 M) of M(NO_3_)*_n_* (M*^n^*^+^ = Ag^+^, Al^3+^, Ca^2+^, Cd^2+^, Co^2+^, Cr^3+^, Cu^2+^, K^+^, Na^+^, Mg^2+^, Mn^2+^, Ni^2+^, and Pb^2+^) were prepared for fluorescence sensing measurements.

## 4. Conclusions

In summary, Cd(II) based coordination polymers **1** and **2** with high water and thermal stability have been hydro(solvo)thermally synthesized. Compound **1** has a non-interpenetrating pillared-bilayer open framework of a topological (4,6,8)-connected net, and compound **2** features a two-fold interpenetrating bipillared-layer condense framework of a topological 6-connected **pcu** net. Compounds **1** and **2** both emit blue light fluorescence and are capable of fluorescence-responsive detection. In terms of metal ion sensing, compounds **1** and **2** both exhibit a remarkable fluorescence quenching effect toward Ag^+^ ions, with excellent selectivity and sensitivity and very low detection limits. In addition, selective and sensitive detection of Al^3+^ and Cr^3+^ ions with low detection limits by **1** and **2** are also achieved through the fluorescence enhancement response.

## Data Availability

Data is contained within the article or Appendix A.

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
