# Peer review of "Fluorescence-Responsive Detection of Ag(I), Al(III), and Cr(III) Ions Using Cd(II) Based Pillared-Layer Frameworks"

_ijms, 2022, doi:10.3390/ijms24010369_

Round 1

Reviewer 1 Report

Wu and co-workers report the synthesis and luminescence properties of two new Cd(II) metal-organic frameworks.  In my opinion, the work is well presented and can be accepted for publication after some revisions:

The synthesis of 2 should be corrected, as it now appears identical to the synthesis of 1.

The differences in the IR spectrum of 2 after treatment with Al and Cr could give very important information on the exact sensing mechanism. Therefore a fully detailed IR analysis and peak assignment should be performed.

The authors could expand a bit more on the relationship between the structures of 1 and 2 and their Al/Cr fluorescence enhancement. Why does 2 shows stronger framework-metal interactions and therefore better enhancement effect?

CheckCif for structure 1 contains an Alert B for the presence of very high residual electron density.
Ideally, the authors should include a response for this, especially since the cif file is not available to
see where this peak corresponds.

Author Response

-The synthesis of 2 should be corrected, as it now appears identical to the synthesis of 1.

Response: This is our mistake. The used reagent has been corrected as naphthalene-1,4-dicarboxylic acid (1,4-H2ndc) for  synthesis of 2.  

-The differences in the IR spectrum of 2 after treatment with Al and Cr could give very important information on the exact sensing mechanism. Therefore a fully detailed IR analysis and peak assignment should be performed.

Response: The IR spectra of 2 after treatment with Al3+ and Cr3+ have been fully characterized as follow, “The antisymmetric and symmetric stretching vibrations of carboxylate group (RCOO) at 1550 and 1360 cm1, respectively, in 2 were largely reduced in intensity after treatment with Al3+ and Cr3+. The weak stretching vibration of benzothiadiazole C=N bond at 1704 cm1 was red-shifted to 1690 cm1 as a strong peak in both Al3+- and Cr3+-treated 2. In addition, the band at 1412 cm1, corresponding to the stretching vibration of benzothiadiazole C=C bond, was slightly blue-shifted by 11 cm1. The differences in the IR spectra of 2 after treatment with Al3+ and Cr3+ in H2O pointed to the framework−M3+ (M3+ = Al3+, Cr3+) interactions that were mainly occupied at the benzothiadiazole N positions and, in part, at the 1,4-ndc2 O positions.”

-The authors could expand a bit more on the relationship between the structures of 1 and 2 and their Al/Cr fluorescence enhancement. Why does 2 shows stronger framework-metal interactions and therefore better enhancement effect?

Response: A bit of discussion on the relationship between the structures of 1 and 2 and their Al/Cr fluorescence enhancement has been addressed as follow, “Structural analyses showed that 1 has large pores of about 25.1% while 2 possesses very small pores of about 0.8%. The difference in pore size makes metal ions passing through the framework more easily in 1 compared to 2, resulting in weaker framework−M3+ (M3+ = Al3+, Cr3+) interactions. In addition, the benzothiadiazole and the 1,4-ndc2 in 2 are close to each other in the pores, creating a comfortable region for binding of metal ion analyte to form stronger framework−M3+ (M3+ = Al3+, Cr3+) interactions, and such an environment is not observed in 1 as the benzothiadiazole and the btc3 are not close to each other.”

- CheckCif for structure 1 contains an Alert B for the presence of very high residual electron density. Ideally, the authors should include a response for this, especially since the cif file is not available to see where this peak corresponds.

Response: A response for the Alert B for the presence of very high residual electron density in the CheckCif for structure 1 has been commented in re-deposit cif file as follow, “The high residual electron density is due to that the overall quality of the diffraction data may be poor, leading to spurious peaks and holes of residual electron density, even the SQUEEZE algorithm was applied.”

Reviewer 2 Report

The manuscript by Wu et al. describes two new Cd based coordination polymers demonstrated sensing for Ag+, Al3+ and Cr3+. The MS is suitable for publication in IJMS but some strong points should be addressed before the acceptance.

1) B-type alerts for cpd1 should be treated or commented.

2) Suspensions of the reported compounds in water should be characterized and discussed in the main text.

3) After water treatment of 1 peaks at 21 and 25 2Theta degrees were splitted. What is the reason of this? Some comments should be added to the main text.

4) Sensing experiments were performed for 1mg of each complex in 3 mL of water. Some evidences should be added that under these conditions both complexes are stable. Moreover ICP-AES / ICP-MS data should be presented for the aqueous solution after the immersing of complexes.

5) The emission of BTD-bpy, H3btc and 1,4-H2ndc should be mentioned.

6) Some literature data concerning complexation of Cr3+ and Al3+ with BTD-bpy, H3btc and 1,4-H2ndc should be added. Especially PL data should be mentioned.

7) Ag+ should produce insoluble polymeric complexes with H3btc and 1,4-H2ndc. Some literature data should be added concerning.

Author Response

1) B-type alerts for cpd1 should be treated or commented.

Response: A response for the Alert B for the presence of very high residual electron density in the CheckCif for structure 1 has been commented in re-deposit cif file as follow, “The high residual electron density is due to that the overall quality of the diffraction data may be poor, leading to spurious peaks and holes of residual electron density, even the SQUEEZE algorithm was applied.”

2) Suspensions of the reported compounds in water should be characterized and discussed in the main text.

Response: Suspensions of the reported compounds in water have been characterized in particle size and zeta potential of surface charge. Some discussions have been stated in the text as follow, “Sensing experiments were performed for 1 mg of each complex in 3 mL of H2O. The well-prepared suspensions of 1 and 2 in H2O exhibited particle sizes of 372.3±102.9 nm and 362.2±100.2 nm, respectively, suggesting uniformity. In addition, the measured zeta potentials of 1 and 2 are –8.41 and 1.46 mV, respectively, in the natural pH conditions, implying negatively and positively charged crystal surfaces, respectively.”

3) After water treatment of 1 peaks at 21 and 25 2Theta degrees were splitted. What is the reason of this? Some comments should be added to the main text.

Response: The XRPD diffractogram of 1 after water treatment has been re-measured. The so-obtained XRPD patterns were in agreement with the pristine samples and were not split at 21 and 25 Theta degrees.

4) Sensing experiments were performed for 1 mg of each complex in 3 mL of water. Some evidences should be added that under these conditions both complexes are stable. Moreover ICP-AES / ICP-MS data should be presented for the aqueous solution after the immersing of complexes.

Response: The XRPD patterns of 1 and 2 after treated with Al3+ and Cr3+ in water for 1 day showed almost unchanged patterns as that of as-synthesized 1 and 2. This is believed to provide strong support for that 1 and 2 under the sensing conditions both are stable.

5) The emission of BTD-bpy, H3btc and 1,4-H2ndc should be mentioned.

Response: The emission of BTD-bpy, H3btc and 1,4-H2ndc have been mentioned as follow: “The photoluminescent properties of BTD-bpy, H3btc, and 1,4-H2ndc ligands and compounds 1 and 2 were studied at room temperature in solid-state and in H2O (Figure S6−S7). Upon excitation, BTD-bpy, H3btc, and 1,4-H2ndc all showed emission band(s) centered at 477 (λex = 335 nm), 384 (λex = 320 nm), and 476 nm (λex = 365 nm), respectively, in solid-state, and at 448 (λex = 315 nm), 342/397 (λex = 270 nm), and 440 nm (λex = 325 nm), respectively, in H2O.”

6) Some literature data concerning complexation of Cr3+ and Al3+ with BTD-bpy, H3btc and 1,4-H2ndc should be added. Especially PL data should be mentioned.

Response: The emission of BTD-bpy, H3btc and 1,4-H2ndc before and after addition of Ag+, Al3+, and Cr3+ in H2O have been measured and addressed as follow: “The emission of BTD-bpy in H2O was quenched after addition of Ag+ while enhanced after addition of Al3+ and Cr3+ ions (Figure S9a); the changes were very close to those of 1 and 2 with the addition of Ag+, Al3+, and Cr3+ in H2O. On the other hand, On the other hand, the emission spectra of H3btc and 1,4-H2ndc showed somewhat changes in intensity with the addition of Ag+, Al3+, and Cr3+ in H2O (Figure S9b and Figure S9c). These findings indicate that Ag+, Al3+, and Cr3+ ions favorably formed analyte−sensor interactions with BTD-bpy struts rather than btc3− and 1,4-ndc2− in 1 and 2 during sensing.”

7) Ag+ should produce insoluble polymeric complexes with H3btc and 1,4-H2ndc. Some literature data should be added concerning.

Response: The powdered solids of 1 and 2 after treated with Ag+ for 1 day gave amorphous manner as confirmed by XRPD patterns. This provides strong support that framework collapse caused fluorescence turn-off effect. Therefore, no further spectroscopic measurements were conducted to check whether the insoluble polymeric complexes of (Ag−btc) or (Ag−1,4-ndc) were produced.

Round 2

Reviewer 1 Report

Thanks to the authors for taking the suggestions into account. I believe the manuscript is now suitable for publication.

Reviewer 2 Report

The MS can be published at the current stage.